



# The stable carbon isotope signature of methane produced by saprotrophic fungi

Moritz Schroll[1]*, Frank Keppler[1,2], Markus Greule[1], Christian Eckhardt[3], Holger Zorn[4], Katharina Lenhart[5,6]*

[1]Institute of Earth Sciences, Heidelberg University, Im Neuenheimer Feld 236, 69120 Heidelberg, Germany
[2]Heidelberg Center for the Environment (HCE), Heidelberg University, 69120 Heidelberg, Germany
[3]Departement of Plant Ecology, Justus Liebig University Giessen, IFZ, 26 - 32, 35392 Giessen, Germany
[4]Institute of Food Chemistry & Food Biotechnology, Justus Liebig University Giessen, IFZ, 58, 35392 Giessen, Germany
[5]Centre for Organismal Studies (COS), Im Neuenheimer Feld 230, 69120 Heidelberg, Germany
[6]Bingen University of Applied Sciences, Berlinstraße 109, Bingen 55411, Germany

*Correspondence to*: Moritz Schroll (Moritz.Schroll@geow.uni-heidelberg.de) and Katharina Lenhart (k.lenhart@th-bingen.de)

**Abstract.** Methane ($CH_4$) is the most abundant organic compound in the atmosphere with emissions from many biotic and abiotic sources. Recent studies have shown that $CH_4$ production occurs under aerobic conditions in eukaryotes such as plants, animals, algae and saprotrophic fungi. Saprotrophic fungi play an important role in nutrient recycling in terrestrial ecosystems by their ability to decompose plant litter. Even though the $CH_4$ production by saprotrophic fungi has been reported, so far, no data for stable carbon isotope values of the emitted $CH_4$ ($\delta^{13}C$-$CH_4$ values) is available. In this study we measured the $\delta^{13}C$ values of $CH_4$ and carbon dioxide ($\delta^{13}C$-$CO_2$ values) emitted by the two saprotrophic fungi *Pleurotus sapidus* and *Laetiporus sulphureus* cultivated on three different substrates pine wood, grass and corn, reflecting both $C_3$ and $C_4$ plants with distinguished bulk $\delta^{13}C$ values. Applying keeling plots, we found that the $\delta^{13}C$ source values of $CH_4$ emitted from fungi cover a wide range from -40 mUr to -69 mUr depending on the growth substrate and fungal species. Whilst little apparent carbon isotopic fractionation (in the range of -0.3 mUr to 4.6 mUr) was calculated for $\delta^{13}C$ values of $CO_2$ released from *P. sapidus* and *L. sulphureus* relative to the bulk $\delta^{13}C$ values of the growth substrates, much larger carbon isotopic fractionations (ranging from -22 mUr to -42 mUr) were observed for the formation of $CH_4$. Whilst the two fungal species showed similar $\delta^{13}CH_4$ source values when grown on pine wood, $\delta^{13}CH_4$ source values differed substantially between the two fungal species when grown on grass or corn. We found that $\delta^{13}CH_4$ source values emitted by saprotrophic fungi are highly dependent on the fungal species and the metabolized substrate. They cover a broad range of $\delta^{13}CH_4$ values and overlap with values reported for methanogenic archaea, thermogenic degradation of organic matter and other eukaryotes.



## 1 Introduction

Methane (CH$_4$) is an important greenhouse gas that is emitted by several abiotic sources (e.g. fossil fuel burning and use, biomass burning, geological processes) and biotic sources (e.g. wetlands, agriculture and waste, fresh waters) to the atmosphere (Kirschke et al., 2013; Saunois et al., 2016, 2019). In the past, biotic CH$_4$ production has been attributed exclusively to strictly

anaerobic microorganisms such as methanogens that are ubiquitous in wetlands, rice paddies, landfills and the intestines of termites and ruminants (Kirschke et al., 2013). The discovery of CH$_4$ emissions from dead and living plants under oxic conditions (Keppler et al., 2006, 2009)  paved the way for the search of new biogenic CH$_4$ sources. Since then, several previously unknown CH$_4$ sources were discovered including endothelial cells of rat liver (Boros and Keppler, 2019; Ghyczy et al., 2008), plant cell cultures (Wishkerman et al., 2011), marine algae (Klintzsch et al., 2019; Lenhart et al., 2016), marine

and terrestrial cyanobacteria (Bižić et al., 2020), humans (Keppler et al., 2016) and saprotrophic fungi (Lenhart et al., 2012). Fungi play a central role in ecosystems by decomposing organic matter and thereby recycling formerly bound carbon and nutrients (Grinhut et al., 2007). This process is especially important in forests where fungi are essential for wood decay and therefore have a great impact on the carbon and nitrogen cycles in these environments (Ralph and Catcheside, 2002). White rot fungi are able to decompose the chemically complex structural component lignin, whereas brown rot fungi mainly

metabolize cellulose and hemicellulose (Ten Have and Teunissen, 2001; Leonowicz et al., 1999; Valášková and Baldrian, 2006). Fungi have already been determined to be involved in the CH$_4$ synthesis during wood decay (Beckmann et al., 2011; Mukhin and Voronin, 2007, 2008) by breakdown of large macromolecules to smaller molecules, thereby providing bacteria and methanogenic archaea with their substrate. Elevated levels of CH$_4$ were found in fungally infected wood stems with oxygen concentrations ranging from 1 to 14 % (Hietala et al., 2015). Here, CH$_4$ production was associated with anoxic microsites in

the xylem, indicating that at least part of the CH$_4$ was produced by methanogenic archaea. Nevertheless, Lenhart et al., 2012 demonstrated that basidiomycetes are able to produce CH$_4$ under aerobic conditions without the presence of methanogenic archaea. Therefore, fungi might be a so far underestimated source of CH$_4$ in the global CH$_4$ cycle.

Applications of stable carbon isotopes (expressed as δ$^{13}$C values) have often been used to investigate sources and sinks of CH$_4$ on the global scale (Whiticar, 1993). As different CH$_4$ sources have distinct δ$^{13}$C fingerprints, they might be used to quantify

the individual contributions of various sources regionally and/or globally (Dlugokencky et al., 2011; Hein et al., 1997; Nisbet et al., 2016; Quay et al., 1999; Tyler, 1986; Whiticar, 1999). The short lifetime of CH$_4$ in the atmosphere (range from 9.7 ± 1.5 to 11.2 ± 1.3 years) (Naik et al., 2013; Prather et al., 2012; Voulgarakis et al., 2013) assures that global isotopic patterns represent the average of recent inputs by various sources and allows the quantification of respective source strengths (Mikaloff Fletcher et al., 2004b, 2004a).

Additionally, stable isotopes provide information about the formation processes of CH$_4$. Traditionally, three formation categories of δ$^{13}$C-CH$_4$ values have been classified: biogenic, with typical δ$^{13}$C-CH$_4$ values ranging from ~-55 mUr to -70 mUr, thermogenic (ranging from ~-25 mUr to -55 mUr) and pyrogenic (ranging from ~-13 mUr to -25 mUr) (Kirschke et al., 2013). However, isotopic patterns of recently identified CH$_4$ sources, i.e.  human CH$_4$ emissions (-56 mUr to -95 mUr) (Keppler

et al., 2016), plant derived $CH_4$ (-52 mUr to -69 mUr) (Keppler et al., 2006), and abiotic UV induced $CH_4$ formation by plants

(-52 mUr to -67 mUr) (Vigano et al., 2009) also need to be considered.

In this study we investigated the stable carbon isotope source signatures of $CH_4$ and $CO_2$ released by the two basidiomycetes *Pleurotus sapidus* (white rot fungus) and *Laetiporus sulphureus* (brown rot fungus*)*. Both fungi were cultivated under sterile conditions on three different substrates (pine wood, grass, and corn) with varying bulk $\delta^{13}C$ values. We examined the influence of fungal species and growth substrate on $\delta^{13}C$-$CH_4$ and $\delta^{13}C$-$CO_2$ values and compared the $\delta^{13}C$-$CH_4$ values from the two

fungal species with those of other known sources reported from the literature.

## 2 Material and Methods

### 2.1 Selected fungi

*P. sapidus* (*Pleurotaceae*, DSMZ 8266) and *L. sulphureus* (*Polyporaceae*, DSMZ 1014) were chosen for this experiment because of their capability to emit $CH_4$ (Lenhart et al., 2012), their ecological and physiological characteristics (white and

brown rot fungi) and well-established practical handling under laboratory conditions.

### 2.2 Cultivation of fungi and incubation experiments

Pine wood, grass and corn were selected as growth substrates. Pine wood was chosen to investigate if brown and white rot fungi differ in $\delta^{13}C$-$CH_4$ and $\delta^{13}C$-$CO_2$ values released during wood decay. Therefore, dead pine wood branches were collected from the forest floor and shredded to small wood chips with a length of about 5 cm (Natura 1800L; Glora, Witten, Germany).

The wood chips were dried at 60°C for 48h and stored in a flask (Weck, Hanau, Germany). Grass ($C_3$ plant) and corn ($C_4$ plant) were selected because of their different stable isotope values. As the metabolic pathway for carbon fixation is biochemically different in $C_3$ and $C_4$ plants, plant biomass differs in $\delta^{13}C$ values, which in turn might lead to different $\delta^{13}C$ values of $CH_4$ and $CO_2$ released by fungi. Therefore, typical garden lawn was manually cut, dried at 70 °C, and stored in a flask. The corn substrate consisted of conventional corn flour.

The substrates were autoclaved and filled into 2.7 l flasks (Weck, Hanau, Germany) and inoculated with pure fungal submerged cultures under sterile conditions according to Lenhart et al., 2012. After addition of the fungi, the flasks were closed with lids and a rubber band sealing. To allow gas exchange during the growth time of the fungi (about two weeks), a hole in the centre of every lid was fitted with a cotton stopper. Before the start of the incubation experiments, the flasks were aerated under sterile conditions in order to start the incubation at atmospheric $CH_4$ mixing ratios. Additionally, to seal the flasks airtight the

cotton stoppers were replaced by sterile silicone stoppers (Saint-Gobain Performance Plastics, Charny, France).

For the incubation experiments, *P. sapidus* und *L. sulphureus* were incubated on the three substrates, while substrates were incubated as control treatments. Before the incubation experiments, the substrates were sterilized by autoclaving. The incubation experiments were conducted as three replicates per treatment. The duration of the incubation accounted for up to 40 h. All incubations were conducted at room temperature (23 ± 1.5 °C). At every sampling point, 40 ml air was taken from





the flasks for gas concentration measurements and an additional 40 ml were taken for $\delta^{13}$C-CH$_4$ stable isotope ratio mass spectrometry (IRMS) analysis. The gas samples were taken with airtight 60 ml PE syringes (Plastipak, BD, Franklin Lakes, USA) and transferred into 12 ml evacuated Exetainers (Labco, High Wycombe, UK). Subsequently a volume of atmospheric air equivalent to the volume of the removed sample was added into each flask directly after sampling. Mixing ratios and stable isotope values of CH$_4$ were corrected according to the dilution.

When calculating the fungal CH$_4$ and CO$_2$ production rates, we subtracted substrate derived CH$_4$ and CO$_2$ production rates (determined in the control treatments) from the respective fungi containing samples. Additionally, for the calculation of the fungal production rates only sample points showing a linear increase in CH$_4$ and CO$_2$ were taken into account.

To account for differences in the metabolic activity of the fungi, we additionally measured respiration rates, assuming that metabolic activity correlates with respiration and therefore CO$_2$ emissions of the fungi. Hence, we related fungal derived CH$_4$

emissions to respiration by calculating the CH$_4$ : CO$_2$ emission ratio.

### 2.3 Analysis of CH$_4$ and CO$_2$ via gas-chromatography

Samples were analysed using a gas chromatograph (GC, Bruker Greenhouse Gas Analyser 450-GC) equipped with a flame ionization detector (FID) and an electron capture detector (ECD) for the detection of CH$_4$ and CO$_2$, respectively. The detector temperatures were set at 300 °C (FID) and 350 °C (ECD). Five reference gases were used for calibrating the GC-system. The

reference gases were in the range of 1 parts per million by volume (ppmv) to 21 ppmv and 304 ppmv to 40,000 ppmv for CH$_4$ and CO$_2$, respectively. Gas peaks were integrated using Galaxie software (Varian Inc., Palo Alto, CA, USA).

### 2.4 Definition of δ values and isotope apparent fractionation

In this paper, all stable carbon isotope ratios are expressed in the conventional 'delta' δ notation, meaning the relative difference of the isotope ratio of a substance compared to the standard substance Vienna Peedee Belemnite (V-PDB) (Eq. (1)).

$$\delta^{13}\text{C} = \frac{\left(\frac{^{13}\text{C}}{^{12}\text{C}}\right)_{\text{sample}}}{\left(\frac{^{13}\text{C}}{^{12}\text{C}}\right)_{\text{V-PDB}}} - 1 \qquad (1)$$

The apparent fractionation ($\varepsilon_{app}$) between fungal $\delta^{13}$C-CH$_4$ or $\delta^{13}$C-CO$_2$ values and the $\delta^{13}$C values of the substrates was calculated according to Eq. (2).

$$\varepsilon_{\text{app CH4 or CO2}} = \frac{\left(\delta^{13}\text{C} + 1\right)_{\text{fungal CH}_4\text{ or CO}_2}}{\left(\delta^{13}\text{C} + 1\right)_{\text{substrate}}} - 1 \qquad (2)$$

We follow the proposal of Brand and Coplen, 2012 and use the term 'urey' (Ur) as the isotope delta unit, in order to conform with the guidelines for the International System of Units (SI). Hence, isotope delta values that were formerly given as -70 ‰, are expressed as -70 mUr.



## 2.5 Measurements of $\delta^{13}CH_4$ and $\delta^{13}CO_2$ values

Stable carbon isotope values of $CH_4$ and $CO_2$ were measured using a continuous flow isotope mass spectrometry system (CF-
IRMS). A HP 6890N GC (Agilent, Santa Clara, USA) was linked to a preconcentration unit for $CH_4$ measurements and an
autosampler A200S (CTC Analytics, Zwingen, Switzerland) for $CO_2$ analysis. The GC was equipped with a CP-PoraPLOT Q
capillary column (Varian, Palo Alto, USA) (27,5 m x 0.25 mm i.d., film thickness 8 µm). The GC was operated with an injector
temperature of 200°C, isothermal oven temperature of 30°C, split injection (10:1) and a constant carrier gas flow of 1.8 ml
min$^{-1}$ (methane-free helium). The GC was coupled to a Delta$^{PLUS}$XL isotope ratio mass spectrometer (ThermoQuest Finnigan,
Bremen, Germany) via an oxidation reactor and a GC Combustion III Interface (ThermoQuest Finnigan, Bremen, Germany).
The oxidation reactor was employed with the following properties: ceramic tube ($Al_2O_3$), length 320 mm, 1.0 mm i.d., with
Ni/Pt wires inside activated by oxygen, reactor temperature 960 °C.

For $CH_4$ measurements with the preconcentration unit, headspace gas samples were transferred to an evacuated 40 ml sample
loop. Methane was trapped on Hayesep D, separated from other compounds by the GC and then introduced into the IRMS
system via an open split. The working reference gas was carbon dioxide of high purity (carbon dioxide 4.5, Messer Griesheim,
Frankfurt, Germany) with a known $\delta^{13}C$ value of -23.6 mUr (calibrated at MPI for Biogeochemistry in Jena, Germany). All
$\delta^{13}C$ values were corrected using two working reference gases of high purity carbon dioxide (Isometric instruments, Victoria,
Canada) with $\delta^{13}C$ values of -23.9 ± 0.2 mUr and -54.5 ± 0.2 mUr that were calibrated against IAEA and NIST reference
substances. The normalization of the sample values was done according to Paul et al., 2007.

## 2.6 Bulk isotope analysis of fungal substrates

Stable carbon isotope values of the bulk substrate were measured using an Elemental Analyzer Flash EA 11112 (Thermo
Fischer Scientific, Germany) coupled to a Delta V IRMS (Thermo Fischer Scientific, Germany). Therefore, 0.06 mg of the
substrate were put into a tin cup and combusted in the Elemental Analyzer. The resulting gases are separated in a GC by a CP-
PoraPLOT Q capillary column (Varian, Palo Alto, USA) (27,5 m x 0.25 mm i.d., film thickness 8 µm) and then reach the Delta
V IRMS via a Conflo IV Universal Continuous Flow Interface (Thermo Fischer Scientific, Germany). Isotope values were
corrected using USGS 40 and USGS 41 standards.

## 2.7 Determination of isotopic source signature of $CH_4$ and $CO_2$ applying keeling plots

For the determination of $\delta^{13}C$ source values of $CH_4$ and $CO_2$ the keeling plot method was used (Keeling, 1958; Pataki et al.,
2003) (Eq. (3)):

$$\delta^{13}C_a = c_b(\delta^{13}C_b - \delta^{13}C_s)\left(\frac{1}{c_a}\right) + \delta^{13}C_s \tag{3}$$

where $c_a$ is the mixing ratio of $CH_4/CO_2$ in the headspace, $\delta^{13}C_a$ is the $\delta^{13}C$ value of $CH_4/CO_2$ in the headspace, $c_b$ is the
mixing ratio of background $CH_4/CO_2$, $\delta^{13}C_b$ is the $\delta^{13}C$ value of background $CH_4/CO_2$ and $\delta^{13}C_s$ the $\delta^{13}C$ source value of the





$CH_4/CO_2$. For a more detailed description of the application of keeling plots for determination of $CH_4$ source signature we refer to the study by Keppler et al., 2016.

$\delta^{13}$C-$CH_4$ source signatures were calculated after the keeling plot method for each flask. Results of the keeling plots are then given as the arithmetic mean of the three individual flasks per treatment with standard deviations (n=3).

$\delta^{13}$C-$CH_4$ source signatures of each flask of *P. sapidus* grown on pine and *L. sulphureus* grown on pine were corrected for $CH_4$ emissions and $\delta^{13}$C-$CH_4$ values of the "pine" control samples using the following mass balance approach (Eq. (4)).

$$\delta^{13}C_{\text{fungi corrected}} = \frac{\left(P(CH_4)_{\text{fungi}} * \delta^{13}C_{\text{fungi}}\right) - \left(P(CH_4)_{\text{pine}} * \delta^{13}C_{\text{pine}}\right)}{\left(P(CH_4)_{\text{fungi}} - P(CH_4)_{\text{pine}}\right)} \tag{4}$$

, where $P(CH_4)$ $_{\text{fungi/pine wood}}$ is the $CH_4$ emitted by the fungi or pine wood and $\delta^{13}$C $_{\text{fungi/pinewood}}$ is the $\delta^{13}$C-$CH_4$ source signature of the fungi or pine wood derived from keeling plots. Corrected $\delta^{13}$C-$CH_4$ source values for *P. sapidus* and *L. sulphureus* are given as the arithmetic mean of the three individual flasks per treatment with standard deviations (n=3).

The determination coefficient ($R^2$) of the keeling plots showed values higher than 0.93, except for *P. sapidus* grown on grass ($R^2$=*0.51*).

**2.8 Statistics**

Mixing ratios and production rates of $CH_4$, $CO_2$, $\delta^{13}$C-$CH_4$ and $\delta^{13}$C-$CO_2$ values and $\delta^{13}$C source values are presented as arithmetic mean of three independent replicates with standard deviations (SD; n = 3). The SDs are given with a confidence interval of 1 $\sigma$. Linear regression analysis, arithmetic means and SDs were calculated using Excel (Microsoft Excel for Office 365 MSO). Two-way analysis of variance (ANOVA) and a post hoc test (Fisher least significant difference) (SigmaPlot

12.2.0.45, USA) were carried out to test for "species" and "substrate" related effects on $\delta^{13}$C-$CH_4$ and $\delta^{13}$C-$CO_2$ source values for each treatment. Differences at the p < 0.05 level were referred to as significant.

**3 Results and Discussion**

In this section, we firstly present the results of $CH_4$ and $CO_2$ production from the two fungal species grown on the three different substrates. This includes emission rates of $CH_4$ and $CO_2$ from the control treatments of pine wood, grass and corn as

well as the molar ratio of $CH_4$ and $CO_2$. Secondly, we then present the respective stable isotope values measured for $CH_4$ and $CO_2$ during the incubation experiments and calculate the stable isotope source values of $CH_4$ and $CO_2$ released by the fungi applying keeling plots. We then compare these values with stable carbon isotope values of the bulk organic matter by calculating the apparent fractionation. Finally, we compare $\delta^{13}$C source values of fungal derived $CH_4$ with sources values known for other $CH_4$ sources from the literature.



## 3.1 Release of CH$_4$ and CO$_2$ from *P. sapidus* and *L. sulphureus*

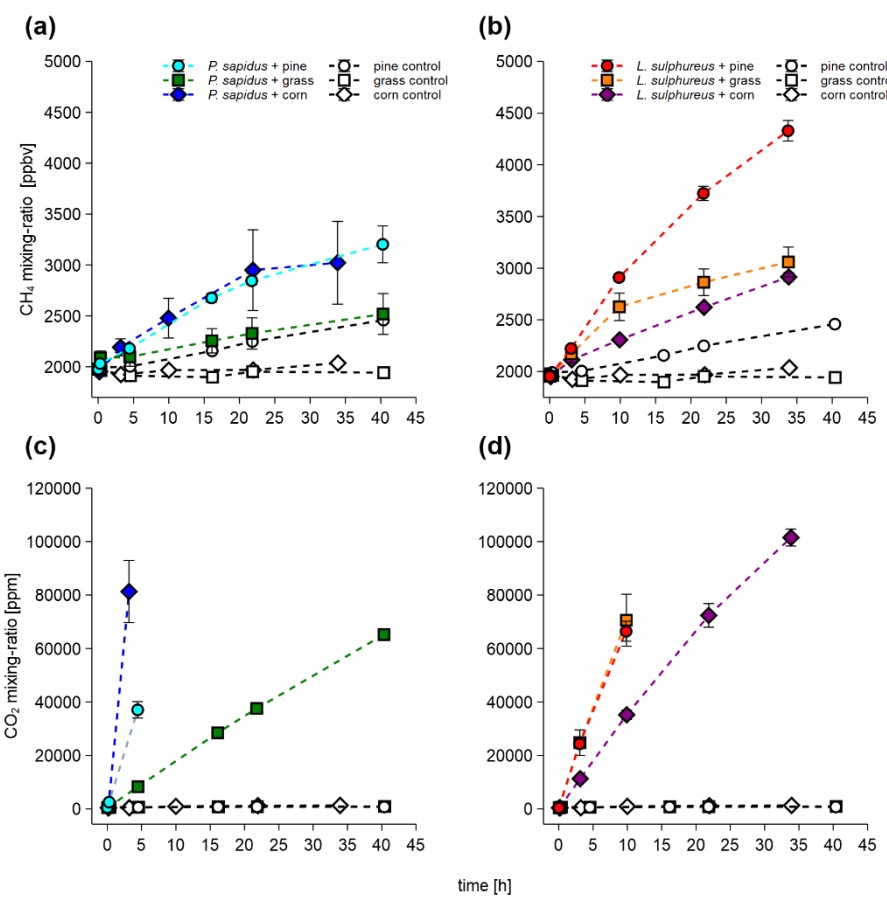

**Figure 1**: Mixing ratios of CH$_4$ and CO$_2$ of *P. sapidus* (a, c) and *L. sulphureus* (b, d) grown on pine wood, grass, and corn. Mixing ratios are presented as mean values with standard deviation SD (n=3).

All incubation experiments where fungi were grown on the different substrates showed a significant increase in CH$_4$ compared to the respective substrate control (Fig. 1 a, c). Calculated emission rates for CH$_4$ and CO$_2$ are presented in Table 1. *L. sulphureus* grown on grass ($7.5 \pm 1.3$ nmol h$^{-1}$) showed the highest emission rate of CH$_4$, followed by *L. sulphureus* grown on pine ($6.2 \pm 0.3$ nmol h$^{-1}$), *P. sapidus* grown on corn ($4.4 \pm 1.9$ nmol h$^{-1}$), *L. sulphureus grown on* corn ($2.6 \pm 0.1$ nmol h$^{-1}$), *P. sapidus* grown on pine ($2.5 \pm 0.2$ nmol h$^{-1}$) and *P. sapidus* grown on grass ($1.4 \pm 0.5$ nmol h$^{-1}$). Please note that CH$_4$ and CO$_2$ emission rates are not related to fungal biomass. Therefore, differences in the emission rates might be due to varying fungal biomass of the subsamples. Instead, CH$_4$ production was related to CO$_2$ production by determining the molar emission ratio between CH$_4$ and CO$_2$ (nmol CH$_4$ : mmol CO$_2$). CO$_2$ production thereby reflects the amount of fungal biomass and is also an indicator for the metabolic activity of the fungi.



Most of the controls did not show significant changes in their $CH_4$ and $CO_2$ mixing ratios over time. However, in the control

flasks of pine wood and corn small $CH_4$ emission rates of $1.3 \pm 0.1$ nmol $h^{-1}$ and $0.25 \pm 0.01$ nmol $h^{-1}$ were observed, and in the control 'grass' the $CH_4$ mixing ratio slightly decreased over time ($-0.05 \pm 0.04$ nmol $h^{-1}$). Whilst the pine wood and corn control flasks showed a small increase in the $CH_4$ mixing ratio, they did not show an increase in $CO_2$ mixing ratios. These data rule out a contamination by microbial heterotrophs, as this would cause a measurable $CO_2$ increase within the flasks. The $CH_4$ increase in the substrate controls might be attributed to $CH_4$ release by dead plant material as it was already shown by Keppler

et al., 2006 and Vigano et al., 2009. Within the scope of these experiments, no analytic test for microbial contamination was conducted. Nevertheless, Lenhart et al., 2012 clearly showed that with the performed method of cultivation of fungi and incubation experiments no methanogenic archaea were present, using three different methods (Fluorescence in situ hybridization (FISH), confocal laser scanning microscopy (CLSM) and quantitative real time PCR). Furthermore, $CH_4$ and $CO_2$ release and the $CH_4 : CO_2$ emission ratios in our incubations are similar to the experiments of Lenhart et al., 2012 and do

not indicate microbial contamination. Therefore, we assume that in our investigations no contamination with bacteria or methanogenic archaea were present.

For *P. sapidus* grown on corn and *L. sulphureus* grown on grass, no further linear increase in $CH_4$ was observed after 22 h and 10 h, respectively. This might be due to a reduced decay of organic matter and slower fungal metabolism because of higher

$CO_2$ and lower $O_2$ mixing ratios.

A drastic increase in $CO_2$ mixing ratios relative to the controls was observed in all flasks containing fungi (Fig. 1 b, d). The $CO_2$ emission rates are shown in Table 1. $CO_2$ production rates ranged from $176 \pm 4$ µmol $h^{-1}$ to $2910 \pm 410$ µmol $h^{-1}$ for *P. sapidus* grown on grass and *P. sapidus* grown on corn, respectively. These highly variable $CO_2$ production rates might reflect different fungal biomass and metabolic activity (mineralisation of organic matter). In the control treatments, tiny increases in

the $CO_2$ mixing ratio were detected ranging from $0.64 \pm 0.12$ µmol $h^{-1}$ to $0.91 \pm 0.14$ µmol $h^{-1}$. Only one flask (corn control) showed a somewhat higher increase in $CO_2$ (7.76 µmol $h^{-1}$), which is most likely caused by microbial contamination of the flask. However, no increase in the $CH_4$ mixing ratio was detected (see supplementary material). Therefore, this control flask was excluded from further calculations.

Mean $CH_4$ and $CO_2$ emission rates and $CH_4 : CO_2$ emission ratios of all treatments are presented in Table 1. Higher ratios

indicate a higher $CH_4$ production during decay of the substrates. Thereby, fungal species and substrate both affect the $CH_4 : CO_2$ emission ratio ($p > 0.001$). For *P. sapidus* $CH_4 : CO_2$ emission ratios are more variable (1.4 to 8.0 nmol $CH_4$/mmol $CO_2$) compared to *L. sulphureus* (6.7 – 9.6 nmol $CH_4$/mmol $CO_2$). This variation might be due to differences in the fungi's enzyme sets required for organic matter decay, as *P. sapidus* is a white rot fungus and *L. sulphureus* is a brown rot fungus. At present the biochemical pathways that lead to $CH_4$ are still unknown, although compounds such as the sulphur-bound methyl-group

of methionine and glucose have been identified to act as carbon precursors of fungal-derived $CH_4$ (Lenhart et al., 2012).

Lenhart et al., 2012 found $CH_4 : CO_2$ ratios of fungi that ranged between 8 nmol $CH_4$/mmol $CO_2$ and 17 nmol $CH_4$/mmol $CO_2$, which is in a good accordance with the $CH_4 : CO_2$ ratios determined in this study. Please note, that $CH_4 : CO_2$ ratios of Lenhart



et al., 2012 were given in ppbv $CH_4$ : % $CO_2$ and for better comparability $CH_4$ : $CO_2$ ratios were converted to fit the units used in this study (nmol $CH_4$ : mmol $CO_2$).


**Table 1:** $CH_4$ and $CO_2$ production rates and molar $CH_4$ : $CO_2$ emission ratios of the fungi incubated on different substrates. Values are presented as mean values of three independent replicates with SD (n = 3), except for the control "corn" (n=2).

| Fungi | Substrate | $CH_4$ production rate [nmol h$^{-1}$] | $CO_2$ production rate [µmol h$^{-1}$] | $CH_4$ : $CO_2$ ratio [nmol/mmol] |
|---|---|---|---|---|
| *P. sapidus* | pine | $2.5 \pm 0.2$ | $901 \pm 79$ | $2.8 \pm 0.4$ |
| | grass | $1.4 \pm 0.5$ | $176 \pm 4$ | $8.0 \pm 2.8$ |
| | corn | $4.4 \pm 1.9$ | $2910 \pm 419$ | $1.4 \pm 0.5$ |
| *L. sulphureus* | pine | $6.2 \pm 0.3$ | $724 \pm 42$ | $8.6 \pm 1.0$ |
| | grass | $7.5 \pm 1.3$ | $771 \pm 103$ | $9.6 \pm 0.5$ |
| | corn | $2.6 \pm 0.1$ | $385 \pm 20$ | $6.7 \pm 0.4$ |
| control | pine | $1.3 \pm 0.1$ | $0.64 \pm 0.12$ | - |
| | grass | $-0.05 \pm 0.04$ | $0.91 \pm 0.14$ | - |
| | corn | $0.25$ | $0.66$ | - |



## 3.2 Stable carbon isotope values of CH$_4$ and CO$_2$

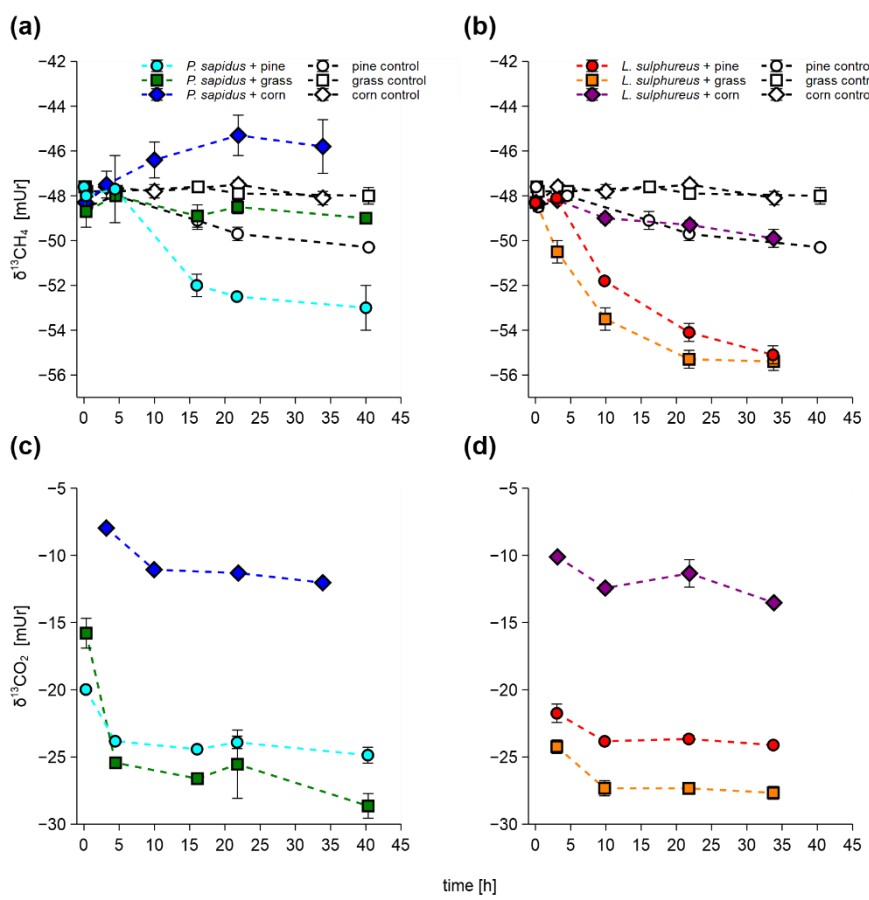

**Figure 2:** Stable carbon isotope values of CH$_4$ and CO$_2$ of *P. sapidus* (a, c) and *L. sulphureus* (b, d) grown on pine, grass, and corn. Values are presented as mean values with SD (n=3), except for $\delta^{13}$CO$_2$ values of *L. sulphureus* grown on corn (n=2).

Stable carbon isotope values of CH$_4$ and CO$_2$ measured from the incubation experiments are presented in Fig. 2. All incubations show a trend towards more negative $\delta^{13}$C-CH$_4$ values (less $^{13}$C) with time except for *P. sapidus* grown on corn, where a tendency towards more positive $\delta^{13}$C-CH$_4$ values was observed (Fig. 2 a, b). During the incubation, $\delta^{13}$C-CH$_4$ values changed from -47.7 ± 0.1 mUr (for incubation of *P. sapidus* grown on pine/grass) and -48.2 ± 0.1 mUr (for incubation of *P. sapidus* grown on corn and *L. sulphureus* grown on pine/grass/corn) to -53.0 ± 0.7 mUr (*P. sapidus* grown on pine), -48.7 ± 0.3 mUr (*P. sapidus* grown on grass), -45.8 ± 1.2 mUr (*P. sapidus* grown on corn), -55.1 ± 0.4 mUr (*L. sulphureus* grown on pine), -55.4 ± 0.4 mUr (*L. sulphureus* grown on grass) and -49.9 ± 0.4 mUr (*L. sulphureus* grown on corn). The controls showed no significant shift in $\delta^{13}$C-CH$_4$ values except for the control "pine", where an increase in the CH$_4$ mixing ratio along with more





negative values of $\delta^{13}C$-$CH_4$ values occurred over time. This was accounted for when calculating the $\delta^{13}C$-$CH_4$ source signatures for *P. sapidus* grown on pine and *L. sulphureus* grown on pine (see materials and methods 2.7).

The $\delta^{13}C$-$CO_2$ values showed a trend towards more negative values within the first three to four hours of incubation (Fig. 2 c, d). After this time only minor changes of the $\delta^{13}C$-$CO_2$ values occurred. Final $\delta^{13}C$-$CO_2$ values of the incubation were -24.9 ±

0.6 mUr (*P. sapidus* grown on pine), -28.6 ± 0.9 mUr (*P. sapidus* grown on grass), -12.0 ± 0.3 mUr (*P. sapidus* grown on corn), -24.1 ± 0.1 mUr (*L. sulphureus* grown on pine), -27.7 ± 0.5 mUr (*L. sulphureus* grown on grass) and -13.0 ± 0.5 mUr (*L. sulphureus* grown on corn).

**Table 2:** Calculated $\delta^{13}C$-$CH_4$ and $\delta^{13}C$-$CO_2$ source signatures, $\delta^{13}C$ values of the substrates, and $\varepsilon_{app\ CH4}$ and $\varepsilon_{app\ CO2}$. Values

are presented as mean values with the SD (n=3).

| Fungi | Substrate | $\delta^{13}C$-$CH_4$ source [mUr] | $\delta^{13}C$-$CO_2$ source [mUr] | $\delta^{13}C$ substrate [mUr] | $\varepsilon_{app\ CH4}$ [mUr] | $\varepsilon_{app\ CO2}$ [mUr] |
|---|---|---|---|---|---|---|
| *P. sapidus* | pine | -65.3 ± 1.1 | -24.1 ± 0.1 | | -38.4 ± 1.2 | 4.0 ± 0.1 |
| | grass | -52.9 ± 1.6 | -27.4 ± 1.3 | | -21.8 ± 1.7 | 4.6 ± 1.3 |
| | corn | -39.8 ± 2.0 | -12.0 ± 0.3 | | -28.5 ± 2.0 | -0.3 ± 0.3 |
| *L. sulphureus* | pine | -61.4 ± 0.5 | -25.0 ± 0.5 | | -34.4 ± 0.6 | 3.0 ± 0.4 |
| | grass | -69.2 ± 1.9 | -29.0 ± 0.5 | | -38.6 ± 2.0 | 2.9 ± 0.5 |
| | corn | -53.4 ± 1.1 | -12.8 ± 0.3 | | -42.2 ± 1.1 | -1.1 ± 0.3 |
| control | pine | | | -28.0 ± 0.5 | | |
| | grass | | | -31.5 ± 0.6 | | |
| | corn | | | -11.7 ± 0.1 | | |





### 3.3 $\delta^{13}$C-CH$_4$ and $\delta^{13}$C-CO$_2$ source signatures of fungi

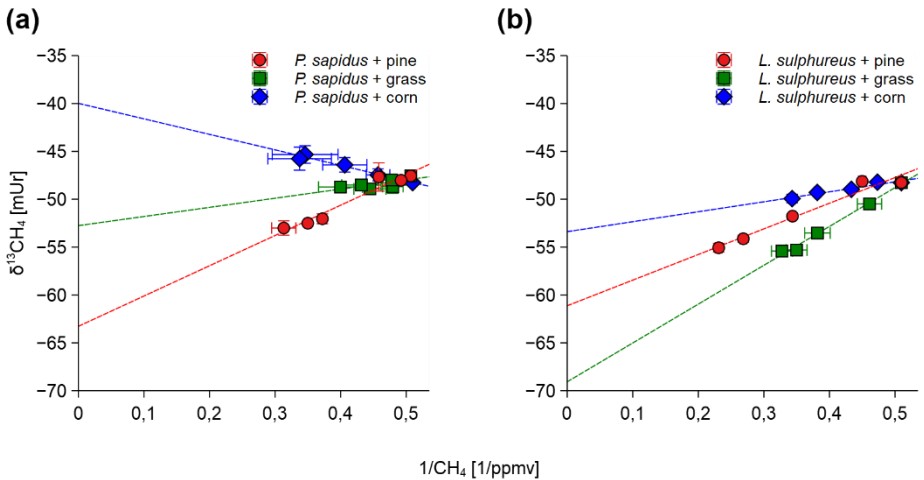

**Figure 3:** Keeling plots shown for *P. sapidus* (a) and *L. sulphureus* (b) grown on three substrates. Sample points in the graphs are given as the arithmetic mean of $\delta^{13}$C-CH$_4$ or $\delta^{13}$C-CO$_2$ values with SD (n=3) on the y-axis and the arithmetic mean of the
inverted mixing ratio of CH$_4$ or CO$_2$ with SD (n=3) on the x-axis.

The $\delta^{13}$C-CH$_4$ source signatures determined via a keeling plot analysis (Fig. 3) are presented in Table 2 and range from -69.2 $\pm$ 1.9 mUr (*L. sulphureus* grown on grass) to -39.8 $\pm$ 2.0 mUr (*P. sapidus* grown on corn). Average $\delta^{13}$C-CH$_4$ source signatures for each fungal species, considering all three substrates, are -52.6 mUr for *P. sapidus* and -61.3 mUr for *L. sulphureus*. These
results suggest that the fungal species significantly influence the isotopic values of the emitted CH$_4$ (*p<0.001*). A possible explanation for this observation could be the different enzyme sets of both fungi decomposing different components of the growth substrates, as *P. sapidus* belongs to white rot fungi and *L. sulphureus* is a brown rot fungus. However, detailed investigations of the metabolic pathways leading to CH$_4$ formation were beyond the scope of this study.

Furthermore, a significant effect of the growth substrate on $\delta^{13}$C-CH$_4$ source signatures was observed (*p<0.001*). $\delta^{13}$C-CH$_4$
source signatures by *P. sapidus* were more positive compared to those of *L. sulphureus* when grown on grass ($\Delta$=16.3 mUr) and corn ($\Delta$=13.6 mUr) (Fig. 4). When grown on pine wood, $\delta^{13}$C-CH$_4$ source signatures were similar with *P. sapidus* showing slightly more negative values ($\Delta$=-3.9 mUr). Methane emitted by both fungi grown on corn was generally more enriched in $^{13}$C (less negative $\delta^{13}$C-CH$_4$ source values) compared to the fungi grown on pine wood and grass. This might be easily explained by the $\delta^{13}$C values of the growth substrates corn (-11.7 mUr, typical for C$_4$-plants) being roughly 20 mUr less
negative in their $\delta^{13}$C values compared to the C$_3$-plants pine wood (-28.0 mUr) and grass (-31.5 mUr).





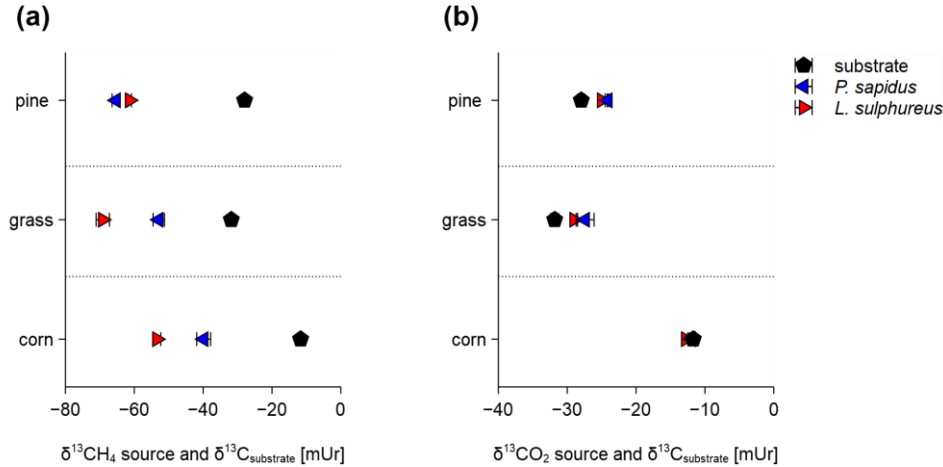

**Figure 4:** Calculated source signatures of $\delta^{13}$C-CH$_4$ values (a) and $\delta^3$C-CO$_2$ values (b) by *P. sapidus*, *L. sulphureus* and the $\delta^{13}$C values of the substrate. Values are presented as mean values of the individual keeling plots with SD (n=3).

Comparison of calculated $\delta^{13}$C-CH$_4$ source signatures with measured bulk $\delta^{13}$C values of the substrates shows that CH$_4$ emitted by both fungi is generally depleted in $^{13}$C compared to the respective substrates (Fig. 4a). Based on this data we further calculated the apparent fractionation ($\varepsilon_{app\,CH4}$) between the $\delta^{13}$C-CH$_4$ source signatures and the bulk $\delta^{13}$C values of the growth substrates. The apparent fractionation was calculated as so far no metabolic pathway for the formation of CH$_4$ in fungi is known and therefore currently only the initial $\delta^{13}$C signatures of the substrates and the calculated $\delta^{13}$C-CH$_4$ source signatures of the

fungi can be compared. The values of $\varepsilon_{app\,CH4}$ are presented in Table 2 and range from -21.8 mUr (*P. sapidus* grown on grass) to -42.2 mUr (*L. sulphureus* grown on corn). When grown on pine wood $\varepsilon_{app\,CH4}$ values are similar for *P. sapidus* (-38.4 ± 1.2 mUr) and *L. sulphureus* (-34.4 ± 0.6 mUr). The differences in $\varepsilon_{app\,CH4}$ values between both fungal species are more distinct when grown on grass (*P. sapidus*: -21.8 ± 1.7 mUr, *L. sulphureus:* -38,6 ± 2.0 mUr) and corn (*P. sapidus*: -28,5 ± 2.0 mUr, *L. sulphureus*: -42.2 ± 1.1 mUr).

The calculated $\delta^{13}$C-CO$_2$ source signatures of both fungi (Table 2) range from -29.0 ± 0.5 mUr (*L. sulphureus* grown on grass) to -12.0 ± 0.3 mUr (*P. sapidus* grown on corn*).* $\delta^{13}$C-CO$_2$ source signatures are in a similar range for both fungi for all three substrates. Although CO$_2$ emitted by *L. sulphureus* is slightly more depleted in $^{13}$C for all three substrates. Hence, an effect of fungal species on the stable carbon isotope values of CO$_2$ is significant (*p=0.008*). Also, the used substrates were found to influence $\delta^{13}$C-CO$_2$ values significantly (*p<0.001*).

The $\delta^{13}$C-CO$_2$ source signatures of the fungi show only small deviations from the bulk $\delta^{13}$C values of the respective substrates (Fig. 4b). However, for both fungi grown on pine wood and grass, $\delta^{13}$C-CO$_2$ values are slightly less negative (a few mUr) compared to the bulk substrate. This observation is rather unexpected, as usually $\delta^{13}$C-CO$_2$ values are more negative with respect to $\delta^{13}$C values of growth substrates due to fractionation during the metabolism (Bowling et al., 2008). However, when





grown on corn $\delta^{13}$C-CO$_2$ source signatures by both fungi are slightly more negative compared to the substrate and calculated

$\varepsilon_{app\ CO2}$ values (Table 2) are -1.1 ± 0.3 mUr and +4.6 ± 1.3 mUr for *L. sulphureus* grown on corn and *P. sapidus* grown on grass, respectively.

The results of the incubation experiments show that there are distinct differences in the patterns of $\delta^{13}$C-CH$_4$ and $\delta^{13}$C-CO$_2$ values released by fungi. While the $\delta^{13}$C-CO$_2$ source signatures are similar to the $\delta^{13}$C values of the substrate (with $\varepsilon_{app\ CO2}$ values up to 4.6 mUr), the $\delta^{13}$C-CH$_4$ source signatures deviate strongly from the respective substrate, with $\varepsilon_{app\ CH4}$ values of up

to -42.2 mUr. This either indicates that metabolic pathways leading to the formation of CH$_4$ and CO$_2$ have different fractionation and/or that fungal CH$_4$ and CO$_2$ derive from different precursor compounds of the respective substrate. The used growth substrates pine wood, grass and corn consist of various components including mainly cellulose, hemicellulose and lignin at different proportions (in contrast if only using pure glucose or cellulose as growth substrate). Hence, the $\delta^{13}$C-CH$_4$ and $\delta^{13}$C-CO$_2$ source signatures might be dependent on specific metabolic pathways of the fungi but also on the chemical

composition of the growth substrate. Therefore, we suggest that the selected fungi and used growth substrates provide a first solid basis for the potential range of $\delta^{13}$C-CH$_4$ values that might occur in nature.

### 3.4 Fungal $\delta^{13}$C-CH$_4$ values compared with known CH$_4$ sources

Figure 5 summarizes $\delta^{13}$C-CH$_4$ values emitted by fungi in relation to other known CH$_4$ sources in the environment that have been reported from the literature. The red bars indicate typical biogenic (formerly only considered to be produced by archaea)

CH$_4$ sources with emissions from wetlands, ruminants, landfills and rice paddies where $\delta^{13}$C-CH$_4$ values are usually ranging from -85 mUr to -40 mUr. Abiotic CH$_4$ sources (including thermogenic or pyrolytic processes) stemming from natural gas, coal mining and biomass burning are characterized by less negative $\delta^{13}$C values usually ranging from -55 mUr to -20 mUr. In addition gas hydrates which might be formed by both microbial and abiotic processes cover a wider range of $\delta^{13}$C values (-29 mUr to -73 mUr), depending on its forming mechanisms (Kvenvolden, 1995). The $\delta^{13}$C source signatures of plant derived CH$_4$

have been reported to be in the range of -72 mUr to -45 mUr (Keppler et al., 2006; Vigano et al., 2009) depending on the three photosynthetic pathways. Furthermore, there is a tendency towards more negative $\delta^{13}$C-CH$_4$ values when the respective plant was treated with UV radiation (Vigano et al., 2009). $\delta^{13}$C-CH$_4$ source signatures of humans which might include both formation by microbes in the gut but also from cellular processes show a rather wide range with values between -95 mUr and -56 mUr (Keppler et al., 2016). The results of our experiments conducted with two fungal species and three different growth substrates

provide a range of $\delta^{13}$C-CH$_4$ source values from -69 mUr to -40 mUr. This range overlaps with other eukaryotic sources, most microbial CH$_4$ sources and even some abiotic CH$_4$ sources such as natural gas or emissions from coal mining.

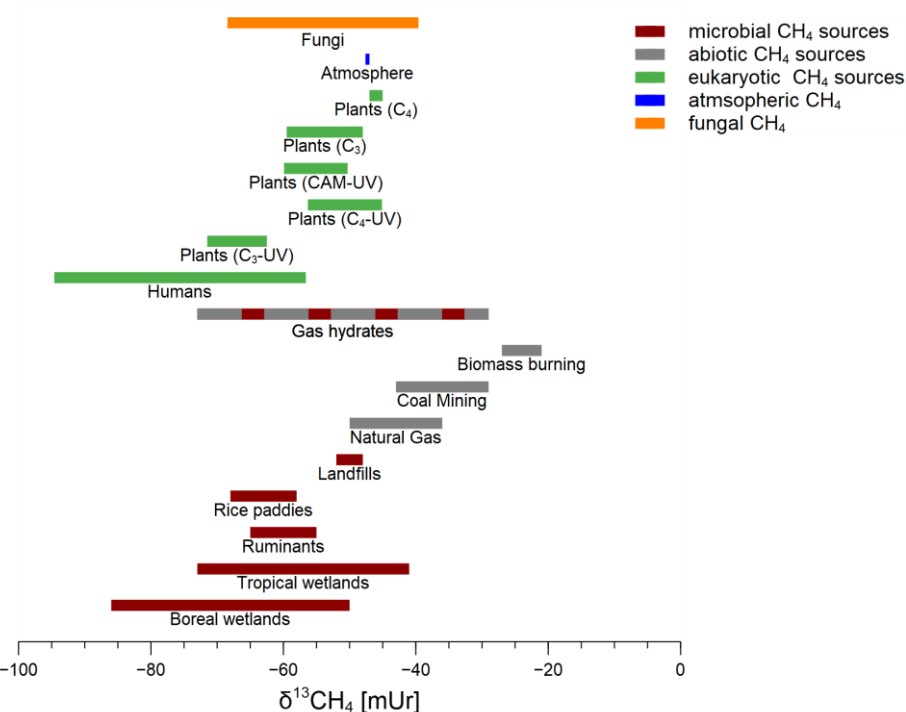

**Figure 5**: Range of δ $^{13}$C-CH$_4$ values of microbial CH$_4$ sources (red), abiotic CH$_4$ sources (grey), eukaryotic CH$_4$ sources (green), atmospheric CH$_4$ (blue) and fungal CH$_4$ from this study (orange). The red and grey dashed bar indicates a mixture of microbial and abiotic CH$_4$ formation processes for gas hydrates (Kvenvolden, 1995). Data taken from (Brownlow et al., 2017; Keppler et al., 2006, 2016; Kvenvolden, 1995; Nisbet et al., 2016; Quay et al., 1999; Vigano et al., 2009).

## 4 Conclusion

This study provided the first analysis of stable carbon isotope values of CH$_4$ emitted by two saprotrophic fungi that were grown on three different substrates. δ$^{13}$C-CH$_4$ and δ$^{13}$C-CO$_2$ source values were found to be dependent on the fungal species, as well as the substrates decomposed by the fungi. δ$^{13}$C-CH$_4$ source values of the fungi were found to be in the range of -69 mUr to -40 mUr and therefore overlap with δ$^{13}$C-CH$_4$ values reported for other CH$_4$ sources such as methanogenic archaea, eukaryotes and from abiotic processes. Stable carbon isotope values of CH$_4$ in combination with flux measurements are often applied for a better understanding of regional and global CH$_4$ cycling. However, in recent years it has become clear that many biogenic CH$_4$ sources include complex CH$_4$ formation processes resulting in different isotopic fractionation patterns depending on several biochemical and abiotic factors. Thus, studying ecosystems in which more than one major CH$_4$ source has to be expected (e.g. methanogenic archaea, fungi, cyanobacteria or plants) distinguishing between each individual source solely by stable carbon isotope values might be highly challenging. Therefore, additional tools are needed to better identify the sources





but also to disentangle sources and sinks. In future research the stable hydrogen isotopic values of $CH_4$ ($\delta^2$H-$CH_4$ values) or even applications of clumped isotopes might prove suitable tools for better distinction between different $CH_4$ sources and thus

to better constrain the global $CH_4$ budget.

*Author contributions.* MS, KL, and FK conceived the study and designed the experiments; HZ provided fungal cultures, MS performed the experiments under the supervision of FK and KL; CE helped with gas measurements; MG measured stable

isotope values of greenhouse gases; MS, FK, and KL analysed the data; MS, FK, HZ, MG, and KL, discussed the results, and MS, KL and FK wrote the paper.

*Competing interests.* The authors declare that they have no conflict of interest.

*Acknowledgments.* We thank Anette Gieseman for analytical measurements of stable carbon isotope values of the bulk substrates. We are grateful to Bianka Daubertshäuser for technical support with the cultivation of the fungi and to Lukas Kohl for encouraging us to perform this study. We acknowledge financial support by the Deutsche Forschungsgemeinschaft.

*Financial support.* This research has been supported by the Deutsche Forschungsgemeinschaft (grant nos. KE 884/8-2, KE

884/16-2) and (LE3381/1-1).







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
