# Peer review of "The stable carbon isotope signature of methane produced by saprotrophic fungi"

_Biogeosciences, 2020_

## Referee Comment (RC1) · Lukas Kohl (Referee) · 16 Apr 2020

Schroll and co-authors studied the stable carbon isotope values of methane emitted during the aerobic decomposition of organic matter by two fungal species. Methane production by fungi during plant litter decomposition is novel pathway of methane formation, that was recently documented by by the authors and others. This manuscript, however, is the first study of the stable carbon isotope (d13C) values associated with this novel pathway and their relationship substrate d13C values.

This study addresses closes a knowledge gap in the isotope systematics of atmospheric methane that is relevant to the Biogeosciences readership. The authors used state of the art methods, and their conclusions are well supported by their results. The

manuscript is well structured and easy to follow.

The study's strength is that this is the first study of its kind and provides unique stable isotope fractionation factors between biomass and methane produced by fungi. The study also used very rebust measurement methods (GC/IRMS with preconcetration) that exceeds the precision, accuracy, and specificity of laser-based analysers. The main limitation of the study are that the authors did not test for contaminations by other microbial species during this study (this was, however, tested by the authors in similar incubations in a previous study). Another limitation is that the authors were not able to identify controls over relatively large variations in methane isotope values beside differences between C3 and C4 plants. This, however, is understandable given that the biochemistryof aerobic methane production in fungi remains poorly understood, and the authors contribution will surely help elucidate these pathways in the future.

Main comments: - The authors used two distinct fungal species, and state that these species include both white rot and brown rot fungi. However, I was unable to find where in the manuscript the authors identify which fungal species belongs to which group.

Minor comments. L53: remove 'applications of' for easier sentence structure L54: clarify what 'they' refers to in 'they might be used..', also, avoid 'fingerprints' ('characteristic d13C values?) L57: 'global isotopic patters': Do you mean the d13C values of atmospheric CH4?' L63: 'isotope patters': stable isotope values? L74f: clarify which fungi is the white rot and which one is the brown rot one. L142: is 0.06mg correct? This seems a very low sample inweight for EA/IRMS, although not impossible. Also, did you analyse analytical replicates? A single 0.06mg inweight is likely associated with a significant subsampling error. L164: You could add a note that the low R2 resulted from the lack of a change in d13C values (emission d13C was similar to background d13C). In this case, a low R2 does not indicate a poor relation between concentration and d13C value. L167: 'The SDs are given with a confidence interval of 1 $\sigma$': sentence not needed and meaningless. L173-179: not needed, can be removed. L185: 'where': use 'in which' instead L229 and Table1: stating CH4:CO2 ratios in umol/mol instead of

nmol/mmol would improve clarity. L302: 'distinct differences in the patterns': redundant structure, could be simplified. L305-311: This section could use some language editing for better flow. e.g. L306: 'the used growth substrates': The growth substrates used for this study... or similar. L306: 'consist of various amounts': contain distinct amount of cellulose, [..], and other compounds. L307: structure in parenthesis: grammar L309: .. source signatures might _depend_ on the metabolic pathyways _used by_ the fungal species _as well as_ the chemical composition of the substrate (or similar) L310: Therefore, we suggest: remove this phrase. "The selected ..." L313: Figure 5 _compares_ _the_ d13C-CH4 values.. L320: 'depending on the photosynthetic pathway (C3, C4, or CAM)'

---

## Referee Comment (RC2) · Anonymous Referee #2 · 17 May 2020

General comments: Methane is the second important anthropogenic greenhouse gas after carbon dioxide. Recent studies have shown that this gas can be produced under aerobic conditions by plants, algae, fungi and animals. In this manuscript, Schroll et al. cultivated two saprotrophic fungi on three different substrates and measured the stable carbon isotope values of methane. This study is the first to report the analysis of stable carbon isotope values of methane emitted from saprotrophic fungi. The authors found that the source values of $\delta 13CH_4$, emitted by the fungi, were dependent on the fungal species and the metabolized substrate. Although this paper has some limitations in terms fungal species and substrates, it certainly opens the door for new and exciting work in the area of aerobic methane emissions. Overall, this is a well-written manuscript and deserves to be published in Biogeosciences after minor revisions.

[Figure]

[Figure]

Specific comments: Abstract L16. eukaryotes, L17-18. ecosystems via decomposition of plant litter L18. Although the methane L19. In this study, L20. The common names of fungi must be mentioned here L21. , cultivated... (pine...), reflecting L21. Which grass? It is better to provide the Latin names of pine, grass (species name) and corn L22. Keeling; K must be uppercase here and in other places L26. 'Whilst' should be replaced; it is mentioned in the previous sentence L28. We found that the values of $\delta$13CH4 emitted L29. What is 'They' in 'They cover'?

Introduction L32. Fossil fuel burning indicates a process but not source; source is fossil fuel, biomass, and... L35. microorganisms, L38. discovered, L43. It is better to delete 'therefore' L43-L44. White rot fungi (e.g., Latin name)... brown rot fungi (e.g., Latin name) L46. in the synthesis of CH4 L48. archaea with essential substrate... in fungus-infected wood stem L52. might be an underestimated L53. It is better to delete 'Applications of'; It is better to start the sentence with Stable isotope procedures L54. 'they' is referred to what? L61. have been identified L64. plant-derived CH4..., and UV-induced CH4... L66. In this study, we...

Material and Methods L73. Pleurotaceae and Polyporacaeae are the family names and should not be italicized. L77. Both common and Latin names should be provided for pine, grass (specific plant species) and corn L92. It is better to provide the temperature for autoclave L109. What are those five different gases? L135. Is 'the working reference gas' the standard reference gas? L143. substrate was put... the resulting gases were separated... L144. 27.5 m ... then reached L148. Keeling L153. Keeling L155. Keeling...Keeling L157. It is better to delete the first 'grown on pine' L163. Keeling L169. Was there a reason for using Fisher test instead of a robust test, such as Tukey's test?

Results and Discussion L177. Keeling L178. The second 'source' can be deleted. L185. 'the' should be deleted. L188. The second 'grown' should not be italicized. L194. Most of the controls? It is better to be specific. L195. ... , respectively were observed L206. was present L220. thereby, both...; the 'both' after substrate should

be deleted. L221. Is it P <0.001? ; a comma should be added after sapidus L227. 'in a good accordance' is not clear, it needs to be rewritten. L227. It should be noted that CH4 L262. It is better to rewrite this sentence, like: ...Keeling plot analysis that range ... are presented. L265 and L269. P <0.001 (number should not be italicized) L278. Keeling; one of the 'values' should be deleted. L283. 'as so far' is not clear L285. The values of... that range from... are presented in Table 2 L287. 'more' should be deleted. L292. 'Although... substrate' is not a sentence and should be rewritten. L295. 'usually' should be deleted from here and added after 'are' L299. 'slightly more' should be reworded. L306. CH4 and CO2 are derived L318. a wide range

Conclusion L336. sources, such as methanogenic archaea and eukaryotes. L337. 'and from abiotic processes' should be deleted or modified in such a way to show sources L339. processes, resulting L340. The sentence that starts with 'Thus, studying' is not clear and should be rewritten. L343. research, stable L359. Grant Numbers L384. In CO2, 2 should be subscript. L391. The title of this paper should be written in correct format. L441. The Latin name should be italicized. L447. In CH4, 4 should be subscript. L448. In CH4, 4 should be subscript; In 13C/12C, 13 and 12 should be superscript. L451-L452. CH4 and 13C/12C should be written in correct format. L512. The Latin name should be italicized. L533. Plant Cell Environ.

---

## Author Comment (AC1) · 5 Jun 2020

We thank Lukas Kohl for the positive evaluation of our work and for the helpful comments to improve the manuscript. All comments and requested changes were taken into account. Please note that comments by the referee are in italics and that in the authors' answer the mentioned line numbers refer to the version of the revised manuscript including track changes.

Referee 1: Schroll and co-authors studied the stable carbon isotope values of methane emitted during the aerobic decomposition of organic matter by two fungal species. Methane production by fungi during plant litter decomposition is a novel pathway of methane formation, that was recently documented by the authors and others. This

manuscript, however, is the first study of the stable carbon isotope (d13C) values associated with this novel pathway and their relationship substrate d13C values. This study addresses/closes a knowledge gap in the isotope systematics of atmospheric methane that is relevant to the Biogeosciences readership. The authors used state of the art methods, and their conclusions are well supported by their results. The manuscript is well structured and easy to follow. The study's strength is that this is the first study of its kind and provides unique stable isotope fractionation factors between biomass and methane produced by fungi. The study also used very robust measurement methods (GC/IRMS with preconcentration) that exceeds the precision, accuracy, and specificity of laser-based analysers. The main limitations of the study are that the authors did not test for contaminations by other microbial species during this study (this was, however, tested by the authors in similar incubations in a previous study). Another limitation is that the authors were not able to identify controls over relatively large variations in methane isotope values beside differences between C3 and C4 plants. This, however, is understandable given that the biochemistry of aerobic methane production in fungi remains poorly understood, and the authors contribution will surely help elucidate these pathways in the future. Authors: We thank the referee for the positive evaluation of our manuscript. The reviewer's concerns are addressed below.

Main comment:

The authors used two distinct fungal species, and state that these species include both white rot and brown rot fungi. However, I was unable to find where in the manuscript the authors identify which fungal species belongs to which group.

Authors: A description of which fungal species belongs to white and brown rot fungi was added to section '2.1 Selected fungi' (L77-78).

Minor comments:

1) L56. remove 'applications of' for easier sentence structure

Authors: Change applied.

2) L57-58. clarify what 'they' refers to in 'they might be used..', also, avoid 'fingerprints' ('characteristic d13C values?)

Authors: 'they' was clarified as '$\delta$13C-CH4 values' and 'fingerprints' was changed to 'characteristic $\delta$13C values'.

3) L61. 'global isotopic patters': Do you mean the d13C values of atmospheric CH4?'

Authors: Correct. For clarification purposes '$\delta$13C-CH4' was added.

4) L66. 'isotope patterns': stable isotope values?

Authors: Change applied.

5) L77-78. clarify which fungi is the white rot and which one is the brown rot one.

Authors: A specification of which fungal species belongs to white and brown rot fungi was added to section '2.1 Selected fungi'.

6) L148. is 0.06mg correct? This seems a very low sample inweight for EA/IRMS, although not impossible. Also, did you analyse analytical replicates? A single 0.06mg inweight is likely associated with a significant subsampling error.

Authors: Yes, the sample weight is correct. Around 0.06 mg of sample was used for the EA/IRMS measurements. Three replicates of each substrate were measured (n=3). Standard deviations for $\delta$13C of the substrates were 0.5 ‰ for pine wood, 0.6 ‰ for grass and 0.1 ‰ for corn.

7) L170-173. You could add a note that the low R2 resulted from the lack of a change in d13C values (emission d13C was similar to background d13C). In this case, a low R2 does not indicate a poor relation between concentration and d13C value.

Authors: Thank you for the very helpful comment. We added a note according to the reviewer's suggestion.

8) L176-177. 'The SDs are given with a confidence interval of 1 $\sigma$': sentence not needed and meaningless.

Authors: Change applied.

9) L182-188. not needed, can be removed.

Authors: Please note, that for better readability we would like to keep this paragraph as it clearly explains the structure of section '3 Results and Discussion' and makes this section easy to follow for the reader.

10) L194. 'where': use 'in which' instead

Authors: Change applied.

11) L201-239 and Table1. stating CH4:CO2 ratios in $\mu$mol/mol instead of nmol/mmol would improve clarity.

Authors: Thank you for your suggestion. The units of the CH4 : CO2 ratios were changed accordingly throughout the whole manuscript.

12) L314-315. 'distinct differences in the patterns': redundant structure, could be simplified.

Authors: We reworded the sentence.

13) L318-324. This section could use some language editing for better flow. e.g. L306: 'the used growth substrates': The growth substrates used for this study... or similar.

Authors: This section was revised for a better flow.

14) L319-320. 'consist of various amounts': contain distinct amount of cellulose, [..], and other compounds.

Authors: Changes applied.

15) L320-321. structure in parenthesis: grammar

Authors: Change applied.

16) L321-322. ... source signatures might _depend_ on the metabolic pathways _used by_ the fungal species _as well as_ the chemical composition of the substrate (or similar)

Authors: Change applied.

17) L323. Therefore, we suggest: remove this phrase. "The selected ..."

Authors: Change applied.

18) L326. Figure 5 _compares_ _the_ d13C-CH4 values.

Authors: Change applied.

19) L334. 'depending on the photosynthetic pathway (C3, C4, or CAM)'

Authors: Change applied.

―――――――――――――――――――

---

## Author Comment (AC2) · 5 Jun 2020

We thank Referee 2 for the positive evaluation of our work and for the helpful comments to improve the manuscript. All comments and requested changes were taken into account. Please note that comments by the referee are in italics and that in the authors' answer the mentioned line numbers refer to the version of the revised manuscript including track changes.

Referee 2: General comments: Methane is the second important anthropogenic greenhouse gas after carbon dioxide. Recent studies have shown that this gas can be produced under aerobic conditions by plants, algae, fungi and animals. In this manuscript, Schroll et al. cultivated two saprotrophic fungi on three different substrates and measured the stable carbon isotope values of methane. This study is the first to report the analysis of stable carbon isotope values of methane emitted from saprotrophic fungi. The authors found that the source values of $\delta 13CH4$, emitted by the fungi, were dependent on the fungal species and the metabolized substrate. Although this paper has some limitations in terms fungal species and substrates, it certainly opens the door for new and exciting work in the area of aerobic methane emissions. Overall, this is a well-written manuscript and deserves to be published in Biogeosciences after minor revisions.

Authors: We thank the referee for the positive evaluation of our manuscript. The reviewer's concerns are addressed below.

Specific comments:

1) L16. eukaryotes,

Authors: Change applied.

2) L17-18. ecosystems via decomposition of plant litter

Authors: Change applied.

3) L18. Although the methane

Authors: Change applied.

4) L19. In this study,

Authors: Change applied.

5) L20-21. The common names of fungi must be mentioned here

Authors: The common names of the fungi have been added to the revised manuscript.

6) L21. , cultivated... (pine...), reflecting

Authors: Change applied.

7) L21-22. Which grass? It is better to provide the Latin names of pine, grass (species name) and corn

Authors: The Latin names of the pine, grass and corn species have been added to the revised manuscript.

8) L23. Keeling; K must be uppercase here and in other places

Authors: Change applied.

9) L27. 'Whilst' should be replaced; it is mentioned in the previous sentence

Authors: Change applied.

10) L29. We found that the values of $\delta$13CH4 emitted

Authors: Change applied.

11) L30. What is 'They' in 'They cover'?

Authors: Change applied.

12) L34. Fossil fuel burning indicates a process but not source; source is fossil fuel, biomass, and...

Authors: Change applied.

13) L37. microorganisms,

Authors: Change applied.

14) L40. discovered,

Authors: Change applied.

15) L45. It is better to delete 'therefore'

Authors: Change applied.

16) L46-47. White rot fungi (e.g., Latin name)... brown rot fungi (e.g., Latin name)

Authors: Examples for white rot fungi and brown rot fungi are now included in the manuscript.

17) L49. in the synthesis of CH4

Authors: Change applied.

18) L51. archaea with essential substrate... in fungus-infected wood stem

Authors: Change applied.

19) L55. might be an underestimated

Authors: Change applied.

20) L56. It is better to delete 'Applications of'; It is better to start the sentence with Stable isotope procedures

Authors: 'Applications of' has been deleted. Please note, that we would like to write 'Stable carbon isotopes', as in this context it refers to stable isotopes in a general meaning.

21) L57-58. 'they' is referred to what?

Authors: Change applied.

22) L64. have been identified

Authors: Change applied.

23) L67-68. plant-derived CH4..., and UV-induced CH4...

Authors: Changes applied.

24) L69. In this study, we...

Authors: Change applied.

25) L76. Pleurotaceae and Polyporacaeae are the family names and should not be italicized.

Authors: Changes applied.

26) L81. Both common and Latin names should be provided for pine, grass (specific plant species) and corn

Authors: Both names have been added to the revised manuscript.

27) L97-98. It is better to provide the temperature for autoclave

Authors: A more detailed description of the autoclave method was added to this section.

28) L114. What are those five different gases?

Authors: The five reference gases were certified gas mixtures of $CH_4$ and $CO_2$ with five different concentrations by Deuste Steininger GmbH. The name of the company was added to the manuscript to clarify the origin of the reference gases.

29) L141-143. Is 'the working reference gas' the standard reference gas?

Authors: We modified 'working reference gas' to read "working standard". We also corrected an error (L142) were the two reference standards are $CH_4$ and not $CO_2$. Those two $CH_4$ reference standards are calibrated and certified and are used for the normalization of the samples. According to the 'Principle of identical treatment' the $CH_4$ reference gases were measured exactly in the same way as the samples.

30) L149. substrate was put... the resulting gases were separated...

Authors: Change applied.

31) L151. 27.5 m ... then reached

Authors: Change applied.

32) L153. Keeling

Authors: Change applied.

33) L159. Keeling

Authors: Change applied.

34) L161. Keeling...Keeling

Authors: Change applied.

35) L163. It is better to delete the first 'grown on pine'

Authors: Change applied.

36) L167. Keeling

Authors: Change applied.

37) L178. Was there a reason for using Fisher test instead of a robust test, such as Tukey's test?

Authors: The statistical evaluation with two way ANOVAs was chosen to conclude if there is a general effect of the fungi and substrates on $CH_4$ and $CO_2$ mixing-ratios, $\delta 13CH_4$ and $\delta 13CO_2$ values and the $CH_4 : CO_2$ emission ratios. The results of the post-hoc tests (Fisher least significance difference and Tukey) are attached in the supplement to this comment. Please note, that the post-hoc tests only have a limited value as there are only three repeated measurements for each parameter (n=3) and post-hoc tests are generally designed for a greater number of repeated measurements. Therefore, we prefer not to show the post-hoc tests in this manuscript and keep the general effects that are expressed by the two-way ANOVAs. Nevertheless, for the $\delta 13C-CH_4$ and $\delta 13C-CO_2$ isotope values p-values calculated with the Fisher LSD and Tukey test are similar and produce only minor differences. Please note that p-values ($> 0.05$) for $CH_4$ and $CO_2$ mixing-ratios might occur because either the quantity of

emitted CH4/CO2 by the fungi is similar and/or the biomass of the fungi within the flasks varies. The manuscript was changed accordingly (L177-178) to clarify that the statistical methods applied in the manuscript refer to the results of two-way ANOVAs.

38) L185. Keeling

Authors: Change applied.

39) L187. The second 'source' can be deleted.

Authors: Change applied.

40) L193. 'the' should be deleted.

Authors: Change applied.

41) L197. The second 'grown' should not be italicized.

Authors: Change applied.

42) L203-205. Most of the controls? It is better to be specific.

Authors: The sentence was modified to be more specific.

43) L205. ... . respectively were observed

Authors: This part of the sentence was replaced because of the changes made to the previous comment 42).

44) L215. was present

Authors: Change applied.

45) L229. thereby. both...; the 'both' after substrate should be deleted.

Authors: Change applied.

46) L230. Is it P <0.001?; a comma should be added after sapidus

Authors: Yes. it is p<0.001! The comma was added after P. sapidus.

47) L237. 'in a good accordance' is not clear. it needs to be rewritten.

Authors: 'in a good accordance' was replaced by 'in the same order of magnitude' to make this sentence clearer.

48) L238. It should be noted that CH4

Authors: Change applied.

49) L272-274. It is better to rewrite this sentence. like: ...Keeling plot analysis that range ... are presented.

Authors: Change applied.

50) L276 and L280. P <0.001 (number should not be italicized)

Authors: Change applied.

51) L289. Keeling; one of the 'values' should be deleted.

Authors: Change applied.

52) L295. 'as so far' is not clear

Authors: 'As so far' was replaced by 'as up to the present' to make the sentence clearer.

53) L297-298. The values of... that range from... are presented in Table 2

Authors: Change applied.

54) L300. 'more' should be deleted.

Authors: Change applied.

55) L304-305. 'Although... substrate' is not a sentence and should be rewritten.

Authors: The sentence was rewritten.

56) L309. 'usually' should be deleted from here and added after 'are'

Authors: Change applied.

57) L311. 'slightly more' should be reworded.

Authors: Change applied.

58) L318. CH4 and CO2 are derived

Authors: Change applied.

59) L331. a wide range

Authors: Change applied.

60) L351. sources. such as methanogenic archaea and eukaryotes.

Authors: Thanks for the note. We changed 'abiotic processes' to 'abiotic CH4 sources' because the term 'abiotic processes' might be misleading. Nevertheless. we would like to keep the 'abiotic CH4 sources' in this sentence.

61) L351. 'and from abiotic processes' should be deleted or modified in such a way to show sources Authors: Please see response to previous comment 60).

62) L353. processes. resulting

Authors: Change applied.

63) L354-357. The sentence that starts with 'Thus. studying' is not clear and should be rewritten.

Authors: Change applied.

64) L358. research. stable

Authors: Change applied.

65) L374. Grant Numbers

Authors: Change applied.

66) L398. In CO2. 2 should be subscript.

Authors: Change applied.

67) L405. The title of this paper should be written in correct format.

Authors: Change applied.

68) L455. The Latin name should be italicized.

Authors: Change applied.

69) L461. In CH4. 4 should be subscript.

Authors: Change applied.

70) L462-L463. CH4 and 13C/12C should be written in correct format.

Authors: Changes applied.

71) L465-467. In CH4. 4 should be subscript; In 13C/12C. 13 and 12 should be superscript.

Authors: Changes applied.

72) L526. The Latin name should be italicized.

Authors: Change applied.

73) L547. Plant Cell Environ.

Authors: Change applied.

Please also note the supplement to this comment:
https://www.biogeosciences-discuss.net/bg-2020-108/bg-2020-108-AC2-supplement.pdf

**Supplement:**

Supplement to Point-by-point response to the issues raised by Referee 2

| p-values for pairwise multiple comparison procedures | Comparison for factor | Comparison | Fisher LSD | Tukey |
|---|---|---|---|---|
| $\delta^{13}C\text{-}CH_4$ | fungi | *P. sapidus* vs. *L. sulphureus* | <0.001 | <0.001 |
| | substrate | corn vs. pine | <0.001 | <0.001 |
| | | corn vs. grass | <0.001 | <0.001 |
| | | grass vs. pine | *0.049* | 0.113 |
| | substrate within *P. sapidus* | corn vs. pine | <0.001 | <0.001 |
| | | corn vs. grass | <0.001 | <0.001 |
| | | grass vs. pine | <0.001 | <0.001 |
| | substrate within *L. sulphureus* | corn vs. pine | <0.001 | <0.001 |
| | | corn vs. grass | <0.001 | <0.001 |
| | | grass vs. pine | <0.001 | <0.001 |
| | fungi within pine wood | *P. sapidus* vs. *L. sulphureus* | *0.021* | *0.022* |
| | fungi within grass | *P. sapidus* vs. *L. sulphureus* | <0.001 | <0.001 |
| | fungi within corn | *P. sapidus* vs. *L. sulphureus* | <0.001 | <0.001 |
| $\delta^{13}C\text{-}CO_2$ | fungi | *P. sapidus* vs. *L. sulphureus* | 0.008 | 0.009 |
| | substrate | corn vs. pine | <0.001 | <0.001 |
| | | corn vs. grass | <0.001 | <0.001 |
| | | grass vs. pine | <0.001 | <0.001 |
| | substrate within *P. sapidus* | corn vs. pine | <0.001 | <0.001 |
| | | corn vs. grass | <0.001 | <0.001 |
| | | grass vs. pine | <0.001 | <0.001 |
| | substrate within *L. sulphureus* | corn vs. pine | <0.001 | <0.001 |
| | | corn vs. grass | <0.001 | <0.001 |
| | | grass vs. pine | <0.001 | <0.001 |
| | fungi within pine wood | *P. sapidus* vs. *L. sulphureus* | *0.164* | *0.164* |
| | fungi within grass | *P. sapidus* vs. *L. sulphureus* | 0.020 | 0.020 |
| | fungi within corn | *P. sapidus* vs. *L. sulphureus* | 0.223 | 0.224 |
| $CH_4$ emission rates | fungi | *P. sapidus* vs. *L. sulphureus* | 0.026 | 0.026 |
| | substrate | corn vs. pine | *0.700* | 0.919 |
| | | corn vs. grass | <0.001 | 0.001 |
| | | grass vs. pine | <0.001 | 0.003 |
| | substrate within *P. sapidus* | corn vs. pine | 0.003 | 0.008 |
| | | corn vs. grass | <0.001 | 0.001 |

| | | | | |
|---|---|---|---|---|
| | | grass vs. pine | 0.277 | 0.510 |
| | substrate within *L. sulphureus* | corn vs. pine | 0.008 | 0.020 |
| | | corn vs. grass | 0.093 | 0.203 |
| | | grass vs. pine | <0.001 | <0.001 |
| | fungi within pine wood | *P. sapidus* vs. *L. sulphureus* | <0.001 | <0.001 |
| | fungi within grass | *P. sapidus* vs. *L. sulphureus* | 0.254 | 0.254 |
| | fungi within corn | *P. sapidus* vs. *L. sulphureus* | 0.089 | 0.089 |
| $CO_2$ emission rates | fungi | *P. sapidus* vs. *L. sulphureus* | <0.001 | <0.001 |
| | substrate | corn vs. pine | <0.001 | <0.001 |
| | | corn vs. grass | <0.001 | <0.001 |
| | | grass vs. pine | <0.001 | 0.003 |
| | substrate within *P. sapidus* | corn vs. pine | 0.346 | 0.602 |
| | | corn vs. grass | 0.010 | 0.026 |
| | | grass vs. pine | 0.002 | 0.005 |
| | substrate within *L. sulphureus* | corn vs. pine | <0.001 | <0.001 |
| | | corn vs. grass | <0.001 | <0.001 |
| | | grass vs. pine | 0.054 | 0.123 |
| | fungi within pine wood | *P. sapidus* vs. *L. sulphureus* | 0.482 | 0.483 |
| | fungi within grass | *P. sapidus* vs. *L. sulphureus* | 0.267 | 0.267 |
| | fungi within corn | *P. sapidus* vs. *L. sulphureus* | <0.001 | <0.001 |
| $CH_4 : CO_2$ emission ratio | fungi | *P. sapidus* vs. *L. sulphureus* | 0.474 | 0.474 |
| | substrate | corn vs. pine | 0.190 | 0.378 |
| | | corn vs. grass | 0.023 | 0.057 |
| | | grass vs. pine | 0.249 | 0.470 |
| | substrate within *P. sapidus* | corn vs. pine | <0.001 | 0.002 |
| | | corn vs. grass | 0.618 | 0.867 |
| | | grass vs. pine | 0.001 | 0.004 |
| | substrate within *L. sulphureus* | corn vs. pine | <0.001 | <0.001 |
| | | corn vs. grass | 0.001 | 0.003 |
| | | grass vs. pine | 0.035 | 0.083 |
| | fungi within pine wood | *P. sapidus* vs. *L. sulphureus* | <0.001 | <0.001 |
| | fungi within grass | *P. sapidus* vs. *L. sulphureus* | 0.329 | 0.329 |
| | fungi within corn | *P. sapidus* vs. *L. sulphureus* | <0.001 | <0.001 |